

# A coincidence detection system based on real-time software.

**Sindulfo Ayuso**[1]**, Juan José Blanco**[1,2] **José Medina**[1]**, Raúl Gómez-Herrero**[1,2]**, Oscar García-Población**[1,3]**, Ignacio García Tejedor**[1,3]**.**

 [1]{Castilla-La Mancha Neutron Monitor. Space Research Group. Parque Científico y Tecnológico de Castilla-La Mancha. Avda. Buendía, 11. 19005 Guadalajara, Spain.}

[2]{Physics Department. Space Research Group. Universidad de Alcalá. Ctra. Madrid-Barcelona km 33,6. 28871 Alcalá de Henares. Spain.}

[3]{Computing Engineering Department. Stpace Research Group. Universidad de Alcalá. Ctra. Madrid-Barcelona km 33,6. 28871 Alcalá de Henares. Spain.}

Correspondence to: Sindulfo Ayuso (sindulfo.ayuso@edu.uah.es)

## Abstract

Conventional real-time coincidence systems use electronic circuitry to detect coincident pulses (hardware coincidence). In this work, a new concept of coincidence system based on real-time software (software coincidence) is presented. This system is based on the recurrent supervision of the Analog to Digital Converters status, which is described in detail. A prototype has been designed and built using a low-cost development platform. It has been applied to two different experimental sets for cosmic ray muon detection. Experimental muon measurements recorded simultaneously using conventional hardware coincidence and our software coincidence system have been compared, yielding identical results. These measurements have been also validated using simultaneous neutron monitor observations. This new software coincidence system provides remarkable advantages such as higher simplicity of interconnection and adjusting. Thus, our system replaces, at least, three Nuclear Instrument Modules (NIM) required by conventional coincidence systems, reducing its cost by a factor of 40 and eliminating pulse delay adjustments.



KEYWORDS: Muon telescope, coincidence, muon spectrometers, data acquisition process, cosmic rays.

## 1    Introduction

Cosmic rays (CR) are energetic particles that constantly rain through the Earth's atmosphere. They are the source of a uniform background ionizing radiation. Most of the CR energy reaches the Earth's surface in the form of kinetic energy of relativistic muons, which are secondary products of interactions between highly energetic CR and the nuclei of atmospheric particles (Cecchini, 2012). Muons (μ- and μ+) are particles belonging to the lepton family and they have the same charge (negative and positive, respectively) as that of an electron and 207 times its mass.

Coincidence counting is widely used in experimental particle physics with different purposes such as reducing noise; getting directional information (Karapetyan et al, 2013); reducing the probability of a measurement being triggered by independent, unrelated particles; lessening the probability of independent random background events (Remmen, 2012) or identifying energetic particles in multi-element particle telescopes (see e.g. Muller-Mellin, 1995).

Traditional particle detection systems using coincidence rely on dedicated electronic modules. When real-time operation is not required, alternative approaches based on the analysis of recorded pulse information can be used (e.g. Havelka et al., 2002, Brancaccio et al., 2009). These systems are based on the registration of pulse properties (e.g. amplitude voltages) and their corresponding accurate timestamps. The recorded data are then processed by software in order to obtain the coincidence counting rates. In this work we present a new software-based coincidence system capable of real time operation in ground-based cosmic ray detection systems.

Electronic chains based on the Nuclear Instrumentation Module (NIM) standard (US NIM Committee, 1978) are widely used by many experimental Particle Physics laboratories around the world. Two of the most important advantages of the NIM concepts are flexibility and interchangeability. Although NIM modules cannot communicate with each other through the crate


backplane, some modules, like Analog-to-Digital Converters (ADC), provide their own interface to communicate with external devices. Nowadays, suppliers offer updated replacements that can read data from ADCs and transfer it to a Personal Computer (PC), including analysis and data mining software. Their main disadvantages are high cost and the fact that they are not open-source systems.

In contrast, recent advances in microelectronics have put in the market low cost and small size devices with high performance (Arduino, Raspberry Pi, Beaglebone Black, etc…). Some of them are open-source hardware and run open-source operating systems like Linux, which confer them a great versatility to satisfy different user requirements. Moreover, they usually include many General Purpose Inputs Outputs (GPIO), which are very useful to implement communication protocols with other
electronic devices like one or more ADCs.

     The goals of this work are, firstly, the establishment of the theoretical background and conditions allowing software-based real-time coincidence detection (Sect. 3); secondly, the prototype implementation with a low-cost development platform and minimal and simple hardware and software designs (Sect. 4); thirdly, the validation of operation extracting data from a muon telescope (Sect. 5)
and, finally, the prototype testing in two practical applications (Sect. 6). In addition, we will see how our prototype is able to replace at least three NIM modules used in conventional setup for coincidence detection.

## 2   Experimental setup.

In this section we describe the different elements that have been used in our experiment, mainly two muon detectors and some NIM modules, and how they have been setup to achieve the results presented in this paper.

### 2.1   Muon detectors.

We have used two different muon detection systems based on plastic scintillators. The first device
(henceforth MD1, Fig. 1) consist of a pile of two identical detectors, separated by 8.5 cm. The gap between both detection layers is partially occupied by a 30 cm x 30 cm x 5 cm lead block, which rejects



low-energy particles (Chilingarian, 2007), ensuring that only cosmic muons are detected when coincidence between both detectors is applied. Each detector is an opaque sealed box, with a 30 cm x 30 cm x 3 cm plastic scintillator on the bottom and a photomultiplier tube (PMT) type 53 AVP (Philips, 1959) on the top, which are 18 cm apart. This distance is required to be sure that the PMT observes the whole scintillator surface. The PMT has been properly biased to 1270 V. The upper detector and the lead block can be moved horizontally allowing different fields of view for testing the directionality of muon flux.

The second device (henceforth MD2) is made up by a large area plastic scintillator (100 cm x 100 cm x 5 cm, polyvinyletoluene with 65% anthracene), three PMTs gathering the light emerging through three of four lateral sides and a fourth small Bismuth Germanate (BGO) scintillator (hexagonal prism of 3 cm side and a height of 2 cm) working in coincidence with the other three PMTs. This experimental setup operating in quadruple coincidence selects muon trajectories crossing the BGO, which can be moved over the surface in order to calibrate a position-sensitive detection system currently under development.

## 2.2   NIM amplification chains.

NIM standardization provides users with the ability to interchange modules and the flexibility to reconfigure or expand nuclear counting systems, as their counting applications change or grow. A typical configuration of a single NIM amplification chain with the main modules used in this work can be seen in the Fig. 2. The detector signal is amplified by the preamplifier and amplifier, which also stretches the pulses, making the ADC conversion easier. When the signal amplitude level is between two preconfigured values, the Single Channel Analyzer (SCA) generates a pulse that triggers the ADC conversion when the ADC is working in "Coincidence mode". The Data Acquisition System (DAS) reads and processes this value and, if needed, transfers the data to the PC.

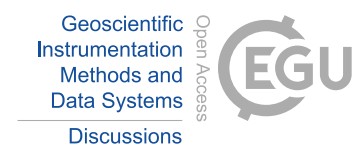

## 2.3  Software-coincidence Acquisition System (SAS).

A new device has been designed and built in order to acquire data from several NIM chains and perform a coincidence policy by means of real-time software, taking advantage of the characteristics of low-cost card-sized embedded-processor platforms. It is also able to store the pulse heights for each separate chain. This device has been validated for our muon detectors based on scintillators with different areas (up to 1 m$^2$). We name this acquisition system SAS (Software-coincidence Acquisition System). Its design and implementation are described in Sect. 4.

## 2.4  ADCs.

The ADC communication protocol is not described in the NIM standard, however, several manufacturers (CANBERRA, FAST, etc.) follow the same basic protocol in their communication lines. In order to understand our software-coincidence basis, the ADC data acquisition process is briefly described below.

The ADC can work in two modes: coincidence mode (COINC) and anticoincidence mode (ANTI). When it works in COINC mode (Fig. 2), input pulse conversions must be enabled or disabled by using the GATE input signal. If the GATE input is low, conversion will not take place.

When it works in ANTI mode, the SCA is not required; the ADC performs peak detection on the signal and provides on its output this maximum as a digital value. The Lower Level Discriminator (LLD) and the Upper Level Discriminator (ULD) potentiometers set the limits for the input signals to be accepted by the ADC for conversion. If an input pulse falls within these limits the ADC starts the conversion process. When the conversion process has finished the Data Ready (DR) signal is activated. When an error occurs in the conversion process the Invalid (INV) line is activated, DR stays inactive and the process aborted (this is important for Sect. 5 discussing). After reading data, the external system (the SAS in our scenario) activates the Data Accepted (DA) line, which resets the ADC, leaving it ready for a new conversion. Once the ADC has started the conversion and up to the DA signal activation, the signal input remains disabled and therefore ignored.



As we will see in Sect. 3.2, the ADC conversion time is needed to perform the software-based coincidence detection code. In this work, the CANBERRA ADC model 8075 has been used and, according to the ADC operator's manual (CANBERRA Industries, 1983), the conversion time is given by Eq. (1):

$$t[\mu s] = 1.5 + 0.01(N + X) \tag{1}$$

Where N is the channel number (quantization) and X is a selected number for 'digital offset' control.

In this work the 'digital offset' control has not been used (X=0) and the channel number has been fixed to N=1024 (10 bits) because higher precision is not needed, so the conversion should always take 11.74 μs.

In order to verify the time before writing the first version of the software, the conversion time with 10 bits of resolution (1024 channels) was experimentally measured with the setup shown in Fig. 3. As it can be seen, the pulse generator output is connected to both the oscilloscope and the ADC, with its output in turn connected to the SAS. When the SAS completes a single reading it asserts the data ready signal. The total conversion time has been determined measuring the time difference between the pulse
from the pulse generator and the pulse from the data ready signal.

The pulse level was adjusted to five different values between 0 and 10 V (minimum and maximum input voltage allowed) and the conversion time was measured for each of them. The results are showed in Fig. 4. As we can see, the higher the pulse height is, the larger the conversion time is. In this work, only minimum and maximum conversion times were needed.

## 3   Coincidence

Particle detection systems are often based in multiple detection layers operating in coincidence. These coincidence-based systems can provide relevant physical information such as particle identity, energy, etc. Relativistic particles, such as high-energy muons, require less than a few nanoseconds to go through
two scintillator layers separated by 1 m (Remmen, 2012). A coincidence in this case is therefore defined by pulses from both scintillators detected within a time window of a few nanoseconds.

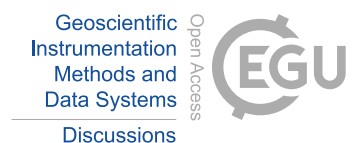

## 3.1 Hardware coincidence

A hardware coincidence circuit is an electronic device with one output and two (or more) inputs. The output is activated only when all input signals are received within a certain time window. Fig. 5 shows a typical coincidence circuit, where the output of the AND gate triggers the ADCs conversion process. This circuit is appropriate for detecting coincidence because of its high speed of operation, since AND gates switching time is only a few nanoseconds (Texas Instruments, 2010).

## 3.2 Software coincidence.

We define software coincidence as the ability to detect coincidence by means of a program running in a CPU-based system. A priori, software-based real time coincidence seems unfeasible because the CPU has to be shared with the underlying operating system. If this is a general purpose operating system, and therefore it has not real-time capabilities, it is impossible to establish deterministically if the acquisition activities will be executed on time. Therefore, the average access time to the hardware has to be taken into account when our software is running. Commercially available embedded development platforms like that used in this work, require a time in the range of microseconds to read and compare two GPIOs (see Sect. 4.2.1 for more detail). This time is several orders of magnitude longer than the time required by a relativistic muon to cross through two stacked detectors. However, if the particle flux is steady and low enough (like that of muons flux), the average time elapsed between the detection of two incident particles will be well above the software processing time, thus allowing for software coincidence processing. This is the basis working principle of our new software coincidence system. As a matter of fact, the total muon flux crossing unit horizontal area from above, at sea-level, is approximately 1 muon per minute per $cm^2$ and remains fairly constant over time (Grieder et al, 2010). Thus, one of our MD1 muon scintillators (30 x 30 cm) would detect about 15 muons $s^{-1}$ (900 muons $min^{-1}$). Given that our telescope is located at 708 m above sea-level, it will detect less than 18 muons per second (Olive et al, 2014). That is, in average, one muon every 55.6 ms and, this period is at least three orders of magnitude above the software processing time (25.2 $\mu$s in the worst case, as will be see further on). This is the basis to demonstrate the feasibility of a software-based coincidence system for low-rate applications.




The process to determine coincidence between two pulses is as follows: the ADC starts the conversion after the leading edge of the input pulse surpasses the LLD setting. If both ADCs receive pulses in coincidence, they will start the conversion at the same time. After finishing the conversion process, each ADC activates their respective DR signals, which are detected by the SAS. This will

decide whether or not coincidence has happened depending on the time elapsed between both DR signals.

As seen in Fig. 4, conversion time depends on the pulse amplitude, so the difference between both conversion times (both DR enables) can be up to 8.8 μs ($16 - 7.2$ μs). In order to solve the coincidence problem, the software is pooling both DR lines continuously. When a DR goes active, the software

waits enough time to be sure that the conversion of the second ADC has finished. Then, it checks the state of the second ADC DR line (DR2). If it has been activated, the software concludes that there is coincidence and the data from both ADC are recorded. After that, the software resets both ADCs sending a DA and the system is again ready for a new event. Otherwise, if DR2 is not active, the software considers that there is no coincidence and sends a DA (reset) to both ADCs without recording

the data. The SAS always sends the DA to both ADC simultaneously (they are both connected at the same circuit wire) to ensure that they are always reset and they start the waiting for a new input pulse at the same time, as suggested by Medina (1987).

Obviously, the received pulses may correspond to different muon arrivals up to 8.8 μs apart and the software could declare them coincident (Fig. 6(a)). Moreover, taking into account the pooling time (4.2

20 μs between each DR, see Table 1 and Sect. 4.2.2 below), particles up to 21.4 μs apart (Fig. 6(b)) could be considered coincident. Actually, in order to guarantee that both ADC conversions have finished, our software waits 25.2 μs (see Sect. 4.2.2). However, in our muon telescope (MD1) it is highly unlikely to have two or more muon arrivals in 25.2 μs because of its steady flux with an average rate of one muon every 55.6 ms. As the arrival of these particles has a random and independent behavior, it follows a

25 Poisson distribution and the probability for detecting k consecutive muons in a δt time window is given by Eq. (2):

$$P(k, \delta t) = \frac{(\lambda \delta t)^k \cdot e^{-\lambda \delta t}}{k!} \tag{2}$$


Where $\lambda$ is the mean number of muons per second.

Considering $\lambda = 18$ muons s$^{-1}$, k = 2 muons and a time window of $\delta t = 25.2$ μs, the probability of having two consecutive muons that would be erroneously counted as coincident is P = 1.03 10$^{-7}$.

Figure 7 shows the Poisson probability for k = 2 muons and a time window $\delta t = 25$ μs according to scintillator area or muon count rate at sea level. As we can see, using scintillators with areas up to 1 m$^2$, the probability of taking as coincident two different muons is negligible. Thus, this software coincidence technique can be applied accepting a minimal number of errors. In order to reduce the number of wrong coincidences as much as possible, the acquisition chains must be adjusted (discrimination levels of ADC, LLD and ULD) in such a way as the particle detected is in the muon energy range, avoiding noise and other particles which would increase the total flux.

As mentioned above, the use of software coincidence is limited by the probability of false coincidence we are willing to accept. For low count rates as those of muons ground-based detectors, our prototype can work with most available scintillators (areas up to 3 m$^2$ with probability of false coincidence = 1.1 10$^{-4}$).

## 4   SAS design and implementation.

The design and implementation of the SAS prototype can be split into two well-differentiated parts: hardware and software.

### 4.1   Hardware.

The hardware implementation involved, firstly, the election of the processing platform, secondly, the design and built of the interface card and, finally, the box assembly.

#### 4.1.1   Hardware platform.

In order to minimize the time employed in design and implementation tasks, we have taken full advantage of the available commercial platforms performances. Nowadays, dozens of card-sized embedded processor systems can be purchased with different input and output possibilities. Beaglebone Black (BBB) was chosen because of the following reasons:



- Great number of GPIO and connection possibilities. We need 37 GPIO in this work.

- High processing power.

- Low power consumption.

- Include micro-SD card slot. It is used to store all processed data.

**4.1.2 Interface card and box.**

An interface card and a box have been designed and implemented (Fig. 8 and Fig. 9) taking into account the technical specifications of BBB and its processor manufacturers (G. Cooley and Texas Instruments, 2013 y 2014). The interface is based on the 74LVC245 tri-state transceiver (IDT, 1999), which provides electrical isolation between the BBB processor and external devices (ADCs), 3.3 V to 5 V voltage level

conversion and buffered signals.

**4.2 SAS software.**

The BBB used in this work (revision B) was delivered with the Angstrom distribution of the Linux operating system. Our software has been developed in C++ and it is compiled in the BBB itself. It performs the following tasks:

- GPIO configuration. To access any external device through GPIO, it must be configured by means of a device structure system called "Device Tree". It has its own language to describe which devices should be made available (Power.Org$^{TM}$, 2011).

- Enabling the transceivers of the interface after booting the system.

- Communicating with the ADCs using their protocol.

- Converting ADC binary data to its decimal value.

- Applying software-based coincidence detection.

- Storing the data in a micro-SD card with a time tag (hour, minute, second and millisecond) and number of registered data per minute.

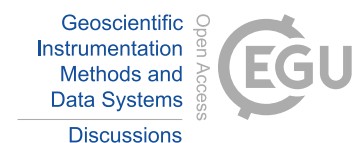

### 4.2.1 Processing time.

After the software development, it was necessary to know the time required to accomplish several tasks, such as reading or writing a bit in a GPIO. As we have seen in Sect. 3.2, to establish the duration of the coincidence time window (in which we consider two pulses as coincident) is a critical decision. Since it must be as short as possible, the software that verifies DR signals must be also as fast as possible. For this reason, after writing code, the execution time of the different routines were verified, revised and optimized to achieve the best results.

In order to measure the time spent in each routine, the software was adapted to make 10 million of iterations of a single task. Thus, the average time of every task was estimated from the total time to run all the iterations (Table 1). In this work, the most relevant task timings are those to get the current status of a single GPIO (4.2 µs) and to set the value of a single GPIO (3.4 µs).

### 4.2.2 Software coincidence detection.

The incident particle causes simultaneous pulses in ADCs inputs and, therefore, both ADCs start the conversion simultaneously. Bearing in mind that software checks DR1 and DR2 sequentially in this order, if DR1 is active and DR2 is inactive in the first checking (Fig. 10 (a)), the minimum waiting time to guarantee ADC2 end conversion (DR2 active) is 4.6 µs, which means another checking of both DR (8.4 µs). Otherwise, if DR2 is active and DR1 is inactive in the first checking (Fig. 10 (b)), the minimun waiting time to ADC1 end conversion is 8.8 µs, therefore the software must wait checking twice both DR (16.8 µs).

Consequently, in order to guarantee that both ADC conversions have finished, the software pool the state of both ADC DR lines three times after each reset, which takes it 25.2 µs. The software considers their state only the third time, if one of both are inactive, it sends the DA signal to reset both ADCs and starting the cycle again (there is no coincidence). Otherwise, if both DR lines are active, a coincidence has been detected and the data is then stored before reseting both ADCs. This process is repeated over and over again.



## 5   SAS experimental validation.

To validate the SAS reliability and proper functioning, several data acquisition experiments with the muon telescope MD1 (see Sect. 2.1) were performed. Both, hardware and software coincidence configurations were simultaneously tested and their results were compared.

Figure 11 shows the experimental setup operating in hardware and software coincidence simultaneously. Two Single Channel Analyzer (SCA) modules and a coincidence detector module are needed in hardware coincidence (see blue block in Fig. 11). In this case, the SAS is used to store data, so it does not work as a coincidence detector and its software is slightly simplified, it waits for the activation of both DR signals to read and store data, and then resets the ADCs. The software

coincidence block (in red) shows how this configuration is significantly simpler than the one based on hardware coincidence, eliminating the need of three modules (two SCA and one coincidence detector).

To make comparisons between both types of coincidence detection systems, we acquired data during one day with the experimental setup shown in Fig. 11. Obviously, the data registered by both software and hardware coincidence chains should be identical. Figure 12 shows the corresponding

histograms produced, which are nearly identical, showing only a minor difference in the total amount of data acquired by both systems (0.05 %).

Although this difference can be considered negligible, further tests were performed in order to find out the origin of this discrepancy. Sometimes, the ADC conversion process produces errors and conversion is aborted (see Sect. 2.4). In these cases, DR signal is not generated and INV signal

activated, which causes data is not registered. Ad-hoc code has been written to register INV signal and several samples has been taken and analyzed. As it can be seen in Fig. 13, an error causes the INV activation in hardware coincidence chain. However the software coincidence prototype stores the correct value because its ADCs have not produced conversion errors. In normal operation, this hardware coincidence data would not be registered and, as a result, the total amount of hardware coincidence data

would be lower than the one registered with software coincidence. That is the origin of the small difference between the histograms corresponding to hardware and software coincidence shown in Fig. 12. Therefore, seeing how the rest of data are similar in time and amplitude values, we conclude that



under the experimental conditions used in this work, both kinds of coincidence detection systems (hardware and software) produce equivalent result.

Furthermore, as the SAS replaces three NIM modules used in the hardware coincidence configuration, simplified connections can be used and, from an economical point of view, the SAS (which cost is about 150 €) can replace laboratory equipment valued over 6 000 €.

## 6    Applications.

The software-based coincidence system presented in this work is an effective low-cost replacement for conventional hardware coincidence, valid for low-rate experimental particle detection systems (up to 500 muons s$^{-1}$ or up to 3 m$^{-2}$ scintillator area at sea level, using our prototype, with probability of false coincidence = 1.1 10$^{-4}$). In this section two examples of specific scientific applications are provided. In the first one, using software coincidence counting greatly reduces the chance of a signal to be caused by an event other than the passage of an energetic muon (Ramesh N. 2011). In the second one, software coincidence is applied to ensure the pulses collected by the PMTs every time correspond to the same passing muon through a small scintillator area.

### 6.1    Monitoring solar activity with ground-based cosmic ray counters.

The muon telescope used in this application (MD1) is installed in the facilities of the Castilla-La Mancha Neutron Monitor (CaLMa) (J. Medina et al, 2013; J.J. Blanco et al, 2014 and O. García-Población et al, 2015). The MD1 and the neutron monitor are located in the same room and their measurements can be directly compared. Neutrons and muons observed at ground level are secondary particles produced by collisions between cosmic rays and atmospheric atoms. The cosmic ray (protons) energy threshold to produce neutrons detected by CaLMa is above 7 GeV because of the geomagnetic location of this neutron monitor, while the energy threshold of primary cosmic rays rises up to higher than 10 GeV for muon production (Duldig, 2000). Transient interplanetary disturbances associated to solar activity can cause decreases in both the neutron and muon count rates observed on Earth surface, in an effect known as Forbush decrease (S.E. Forbush, 1938). In order to observe these cosmic ray flux



variations, the effect atmospheric pressure variations must be removed from the data using a correction procedure (see e.g. Paschalis, 2013 and references therein).

Figure 14 shows pressure-corrected muon and neutron count rates and plasma and interplanetary magnetic field measurements during a Forbush decrease detected by CaLMa on 21 December 2014. The count-rate in CaLMa decreased by 6% with respect the previous neutron count-rate. This decrease was also observed by the muon telescope, working in software coincidence, as a reduction in the steady muon count-rate of about 3%. As can be observed in Fig. 14, a sharp decrease is observed in CaLMa after an interplanetary shock passage that marks the arrival of a complex interplanetary ejecta (21/12/2014 18:00). This complex ejecta seems to be composed by two consecutive Interplanetary Coronal Mass Ejections (ICMEs) and comprises a second interplanetary shock probably related to a compression region created by a fast solar wind stream following the ejecta. The first ICME is listed in the Richardson and Cane ICME list (http://www.srl.caltech.edu/ACE/ASC/DATA/level3/icmetable2.htm) with limits between 22/12/2014 04:00 and 22/12/2014 17:00, including a smooth magnetic field rotation and low and stable solar wind temperature as can be expected when a well-developed magnetic cloud is observed in the solar wind. A second rotation in magnetic field components is observed between 22/12/1014 17:00 and 23/12/2014, suggesting the presence of a second ICME, however solar wind properties show less clear signatures (Hidalgo, 2013).

The good agreement between the muon and neutron data presented in Fig. 14, both of them correlated with the passage of interplanetary disturbances, validates the software-based coincidence system used to acquire the muon data. The difference in the amplitude of the decrease observed by both instruments is likely related to the different energy of the primary cosmic ray producing the secondary neutrons and muons observed at ground level.

## 6.2 Position-sensitive muon detector.

In this experiment we used our software-based coincidence system to acquire data from a prototype of position-sensitive muon detector (MD2, see Sect. 2.1). The experimental setup uses four PMTs





operating in software coincidence. Three of them were placed attached to the sides of a plastic scintillator. The fourth PMT was placed inside an opaque box, gathering the light emitted by a small BGO scintillator (see Fig. 15 (a)). The BGO can be moved horizontally in order to select only muon trajectories crossing certain spot over the surface of the plastic scintillator.

The signal generated by each PMT was amplified and injected to an ADC to carry out its conversion. The four ADCs were connected to our prototype in order to detect coincidence and to record the pulse heights (see block diagram in Fig. 16). The BGO was located in different positions on the big scintillator surface and corresponding data was acquired and registered.

     Figure 15 (b) shows the pulse-height distribution registered by the three lateral PMTs (labelled 1, 2

and 3 in the figure) when the BGO is located over the centre of the plastic scintillator. In this case the three PMTs observed identical distributions.

     Figure 15 (c) shows the pulse-height distribution corresponding to PMT 1 obtained for three different locations of the BGO. As expected, the distribution is shifted towards larger pulse heights when the BGO is located closer to the PMT. The combined pulse height information from PMTs 1, 2

and 3 can be used to reconstruct the location of the particle track.

     Obviously, this practical application could be carried out with hardware coincidence, but we take advantage of the easier adjustment, simpler connection and lower cost of our software coincidence. The final configuration of this application is now under development.

## 7  Conclusions.

A software-coincidence acquisition system (SAS) capable of detecting coincidence by using software and based in a low-cost development platform has been implemented and tested. It works autonomously (i.e. without a dedicated computer) recording data in a micro-SD card and transferring them to a PC through USB or Ethernet connections. In order to evaluate the SAS operation in software coincidence in

comparison with that of hardware coincidence, several tests have been carried out, acquiring and recording data from both coincidence methods simultaneously. The results make evident that software



coincidence is as effective as hardware coincidence with a low flux of particles like that of a cosmic ray ground-based muon telescope (scintillator areas up to 3 m$^2$).

Furthermore, our software coincidence system has been tested in two different experimental setups for cosmic ray muon detection: a two-element muon telescope, requiring single coincidence and a position-sensitive muon detector requiring quadruple coincidence. The results were entirely satisfactory. The first device clearly observed a cosmic ray Forbush decrease, confirmed using neutron monitor data and well correlated with the passage of an interplanetary disturbance. The second device was able to record different PMT pulse levels depending on the location of the incident muon tracks.

This system provides a reliable and low-cost replacement for hardware-based coincidence system modules over forty times its value.

## 8 Acknowledgments

This work has been partially supported by University of Alcalá through the project CCG2014/EXP-013 and by Ministerio de Ciencia y Tecnología through the project ESP2013-48346-C2-1-R.

We would like to thank Mr. José Salvador Pérez Bachiller, who helps us in the SAS box mechanical design, machining and assembly.



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



Table 1. Processing time of some tasks. In yellow background, the task to know the status of a GPIO.

| EXECUTED TASKS | ITERATIONS NUMBER | TOTAL TIME (s) | AVERAGE TIME PER ITERATION (µs) |
|---|---|---|---|
| Setting value of 1 GPIO | $10^7$ | 34 | 3.4 µs |
| Reading only one GPIO | $10^7$ | 42 | 4.2 µs |
| Reading and writing one GPIO | $10^7$ | 87 | 8.7 µs |
| Reading and writing two GPIO | $10^7$ | 162 | 16.2 µs |

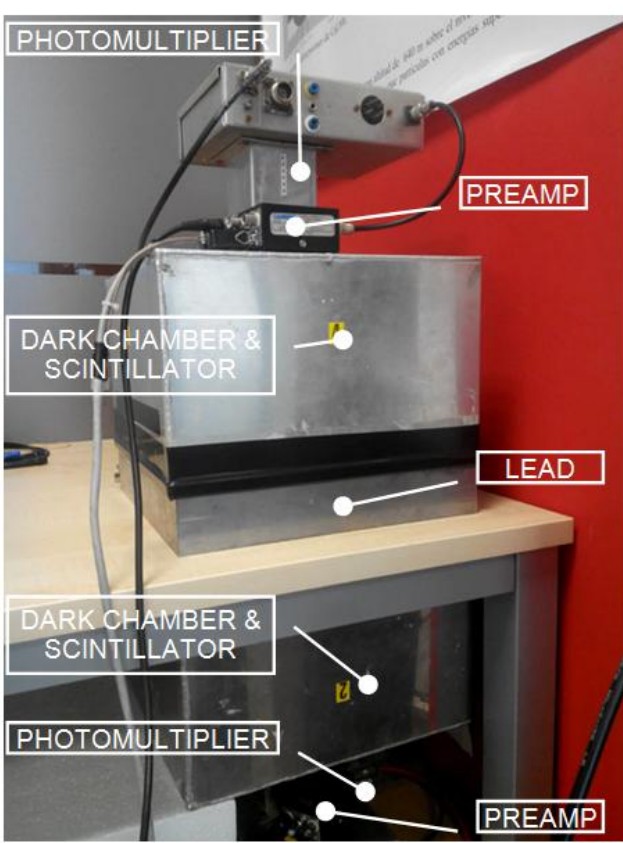

Figure 1. Muon telescope. Main parts.

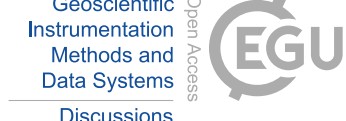



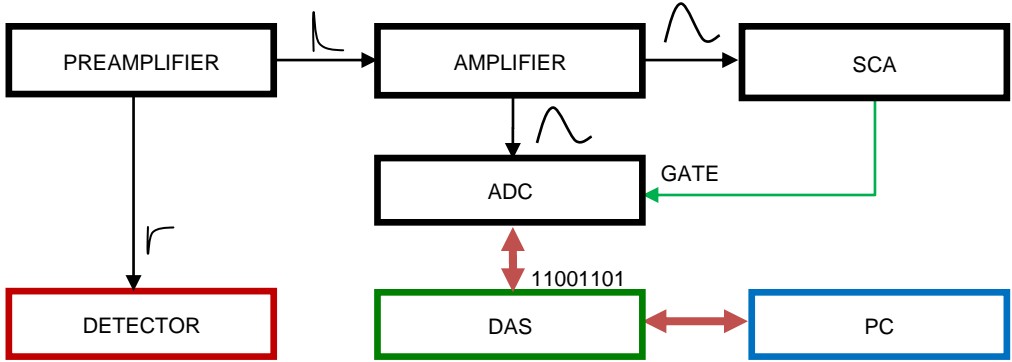

SCA: Single Channel Analyzer.          DAS: Data Acquisition System.

Figure 2. Data acquisition block diagram with NIM modules. ADC working in coincidence mode.

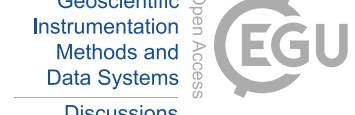



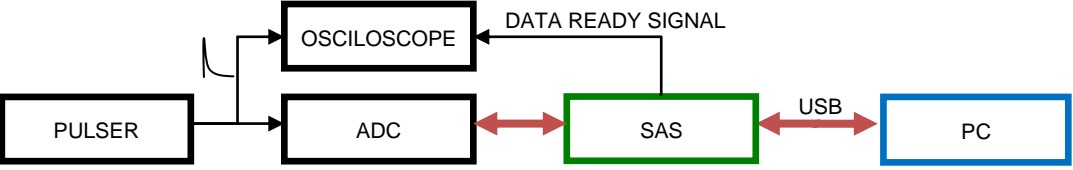

Figure 3. Block diagram used to verify the ADC conversion time. PC is only used to launch and stop SAS software.





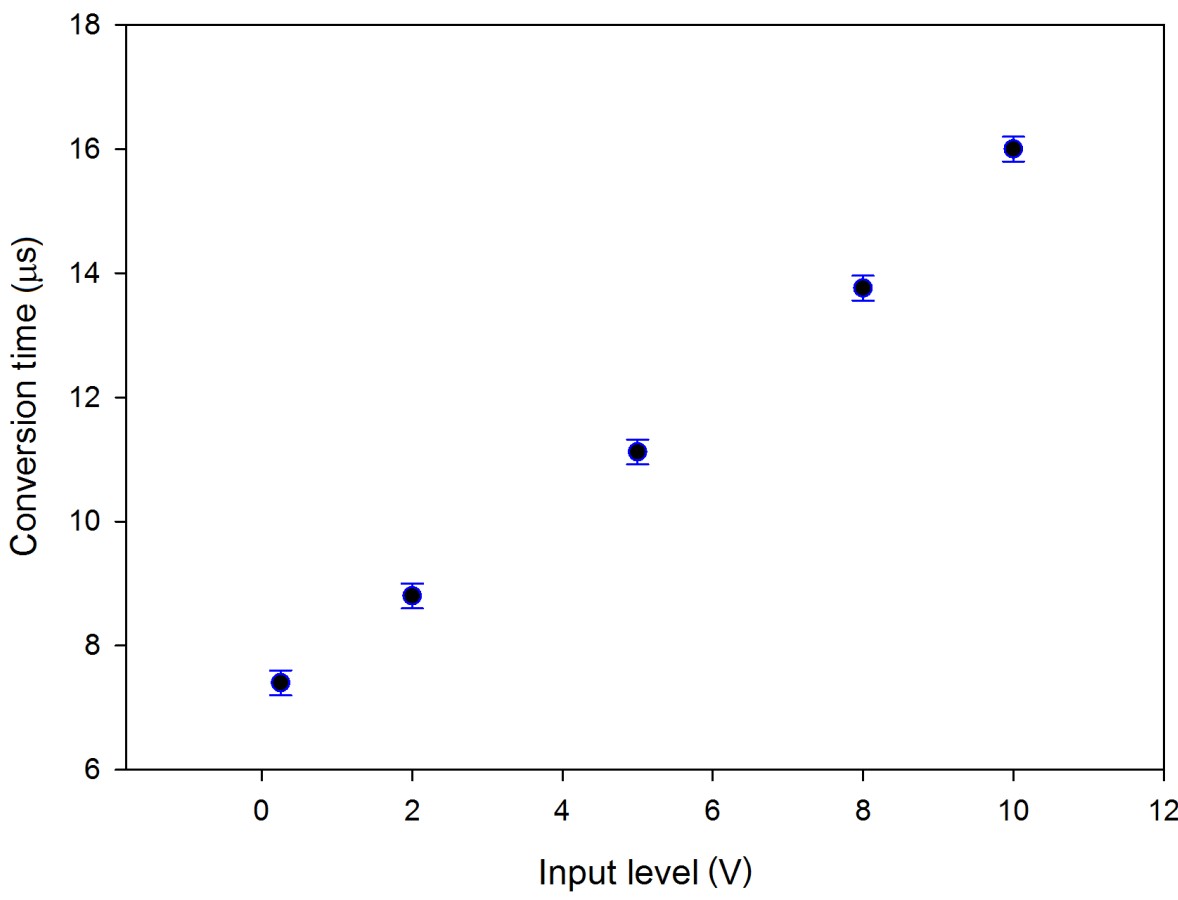

Figure 4. CANBERRA 8075 ADC conversion time. Values between 7.2 and 16 µs with an error rate of ± 0.2µs.




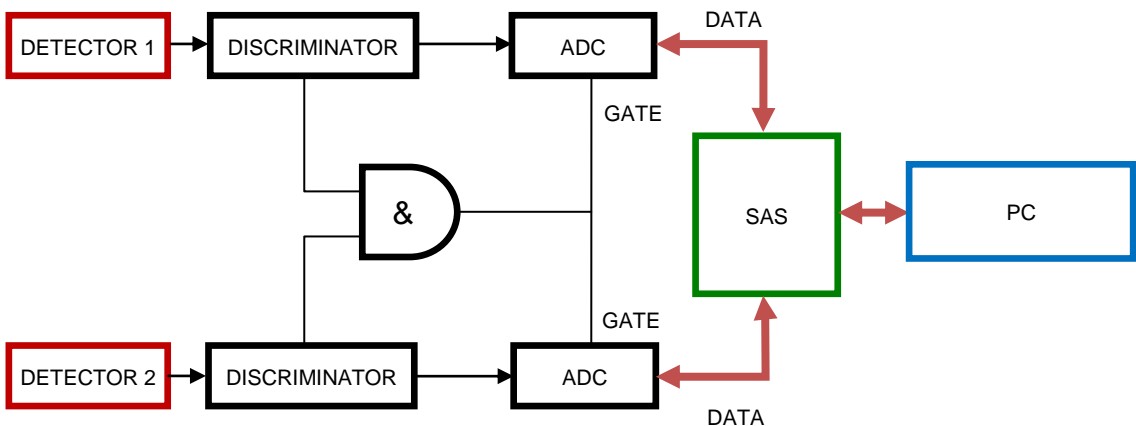

Figure 5. Hardware coincidence detector block diagram. The AND gate is the core of the circuit, its output become active when both inputs are activated. Here, the SAS only acquires and records data. It does not work as a coincidence system.

PC is only used to launch and stop the SAS software.





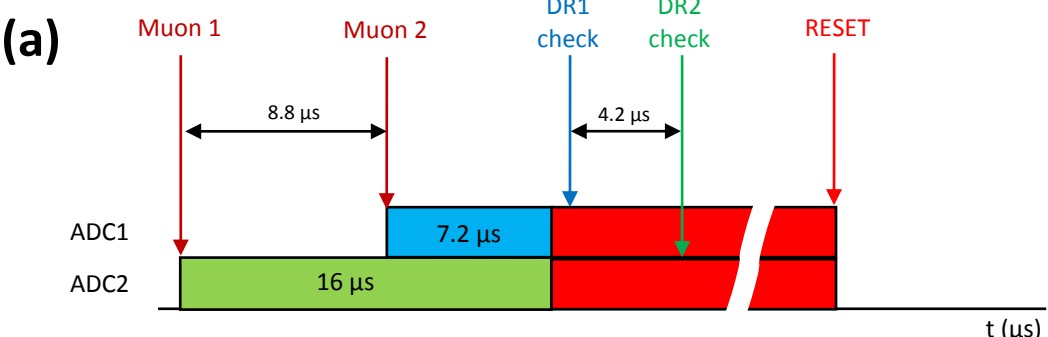

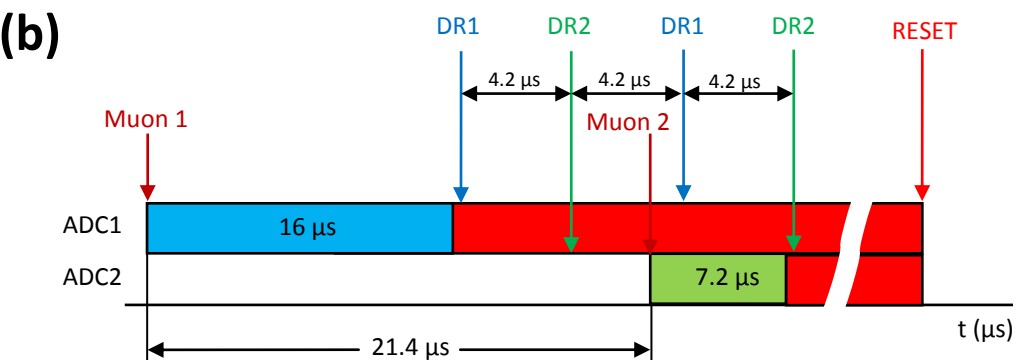

Figure 6. Different muons considered like a single particle (coincident) because of ADC conversion time (blue and green) and software checking time (4.2 µs between DR1 and DR2).

In red, time from the ADC end of conversion to DA activation.

DR1 check: ADC1 Data Ready checking instant.

DR2 check: ADC2 Data Ready checking instant.

(a) Muon 1 generates the maximum height pulse in ADC2 and it takes the maximum conversion time, whereas muon 2 generates the minimum height pulse in ADC1 and it takes the minimum conversion time. Both ADCs finish conversion at the same time and the SAS considers a single muon.

(b) Muon 1: maximum conversion time in ADC1. Muon 2: minimum conversion time in ADC2. In the worst case, taking into account the checking time and the waiting time to ensure the ADC end of conversion, Muon 2 can arrive up to 21.4 µs later than Muon 1 and the SAS considers them as a single muon.





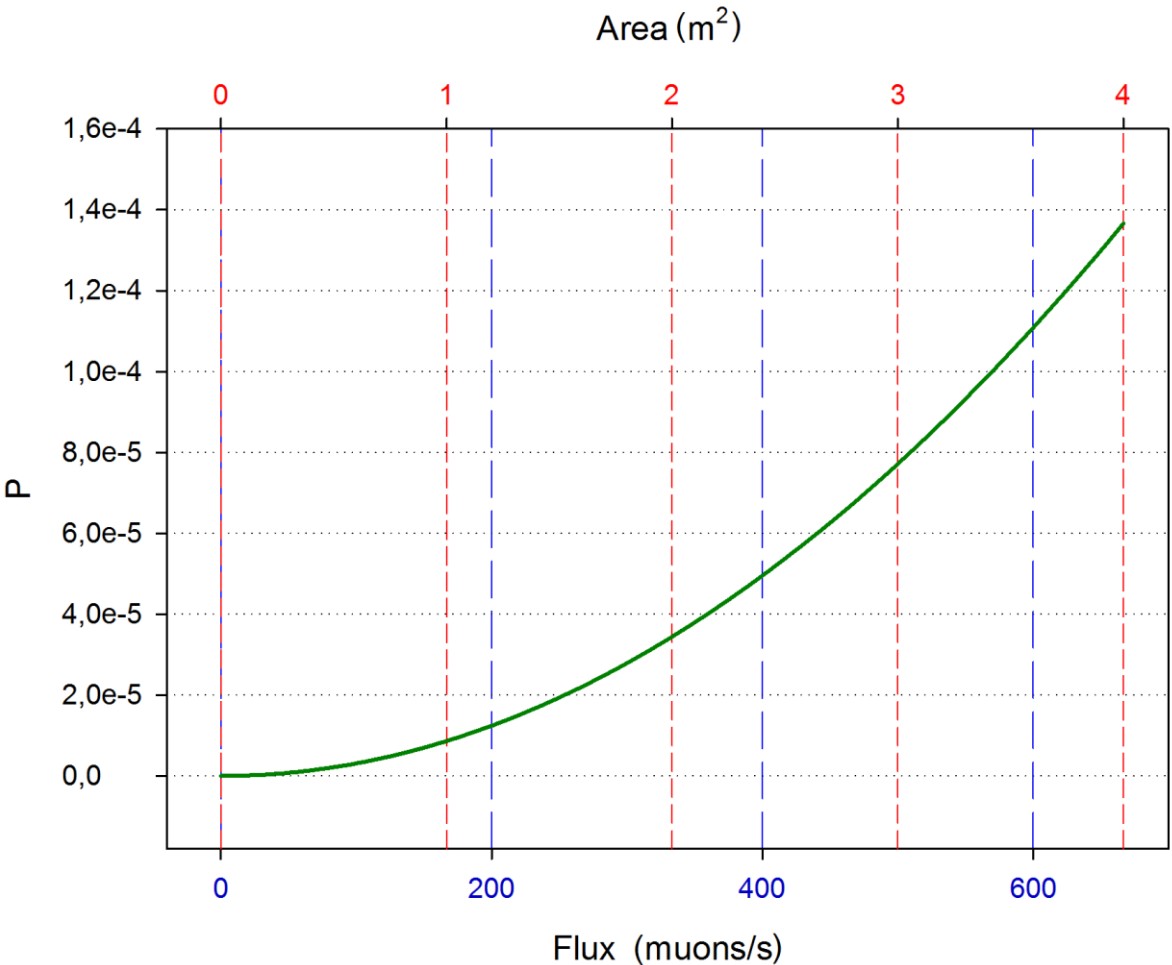

Figure 7. Poisson distribution. Probability of detecting as coincident two different muons in a period of $\delta t=25\mu s$ as a function of muon flux (lower axis) or scintillator area (upper axis) at sea level.

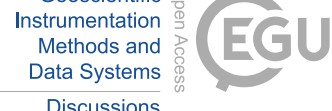

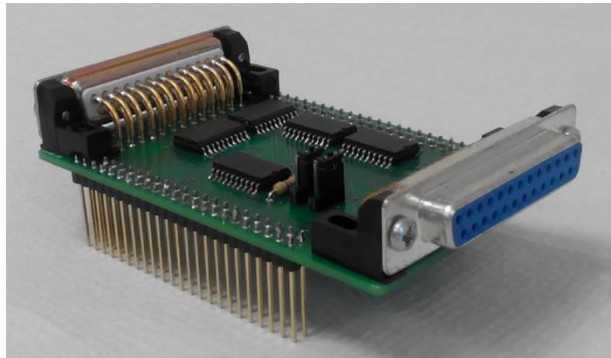
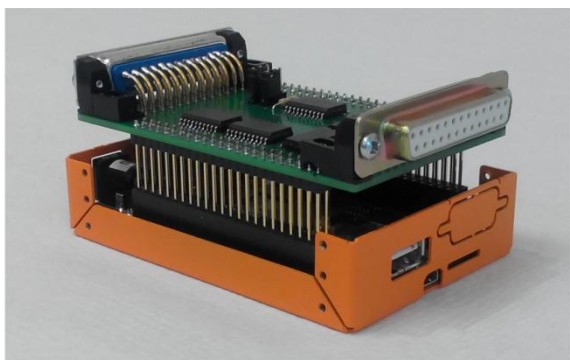

Figure 8. Left: Interface with two DB25 connectors. Right: Interface mounted on the process platform (Beaglebone Black) headers.



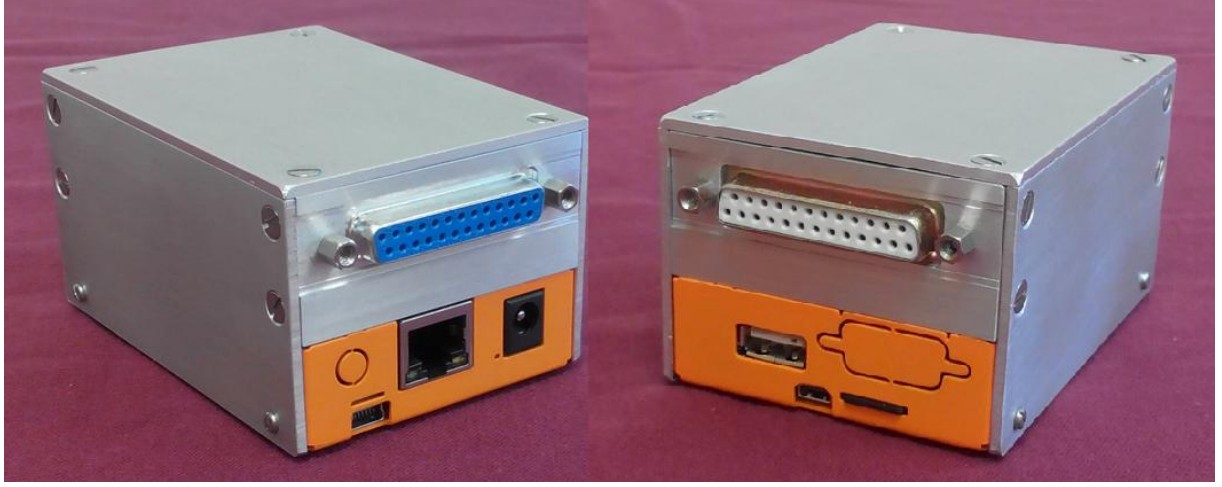

Figure 9. SAS final appearance, sized 90 mm x 60 mm x 48 mm. Both sides feature integrated connectors.



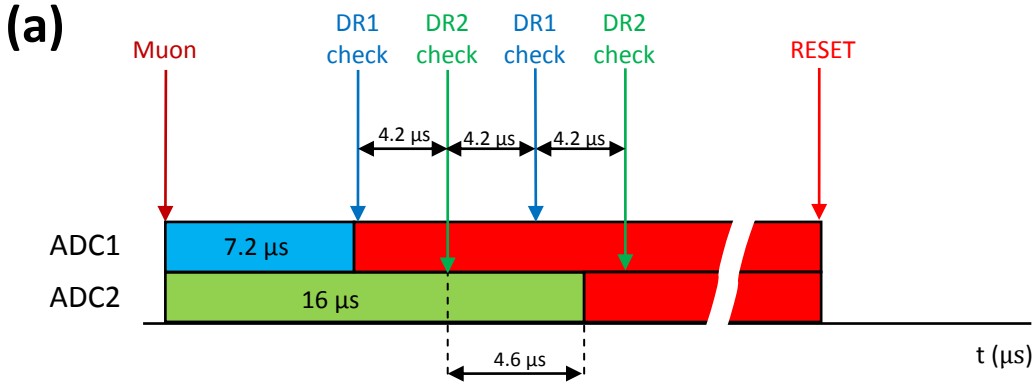

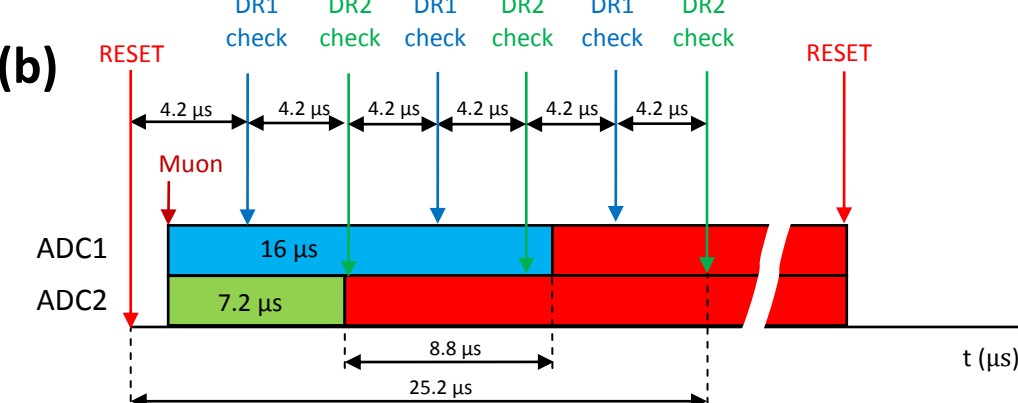

DR1 check: ADC1 Data Ready checking instant.     DR2 check: ADC2 Data Ready checking instant
In red, time from the ADC end of conversion to DA (reset) activation.

Figure 10. Coincidence evaluation. Maximum and minimum conversion time and waiting time to ensure ADC conversion.

(a): ADC1 minimum conversion time and ADC2 maximum conversion time.

(b): ADC2 minimum conversion time and ADC1 maximum conversion time.

In the worst case, after detecting the first ADC end of conversion (b), the SAS must wait checking other twice the DR signal (every time takes 8.4 µs) in order to always ensure the second ADC end of conversion.





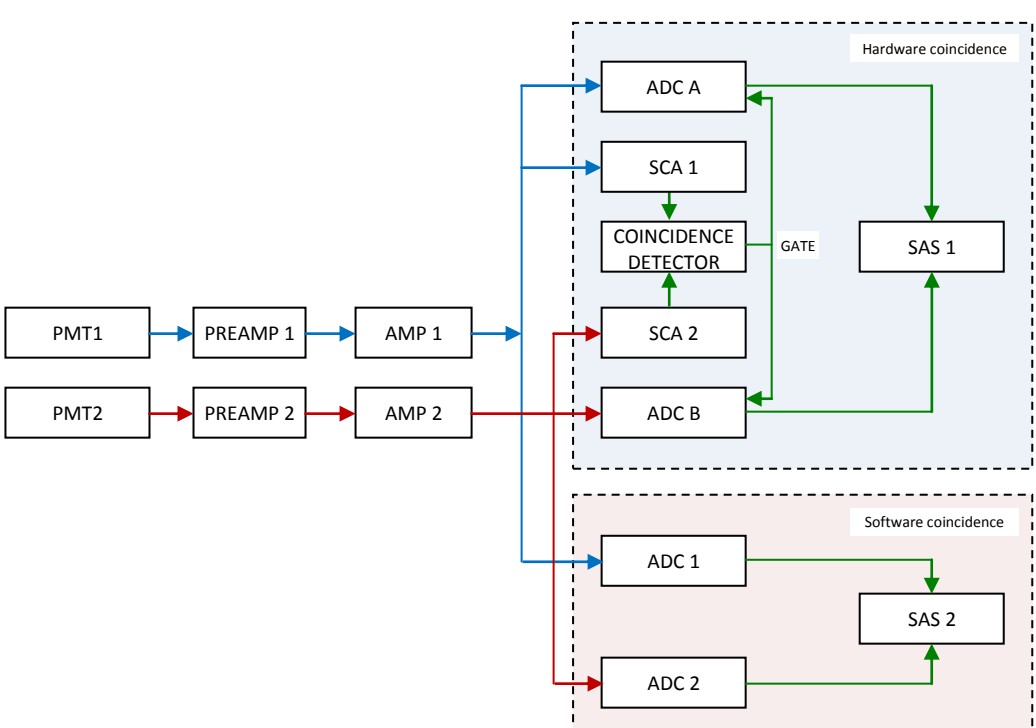

PREAMP: Preamplifier.     AMP: Amplifier.     SCA: Single Channel Analyzer.     PMT: Photomultiplier.

Figure 11. Schematic setup for hardware coincidence and software coincidence results comparison. The
5    same analogue signals detected by PMT1 and PMT2 are introduced into both hardware coincidence and
software coincidence chains. Working in hardware coincidence, SAS 1 only stores data from both
chains. Working in software coincidence, SAS 2 detects coincidence and stores data.



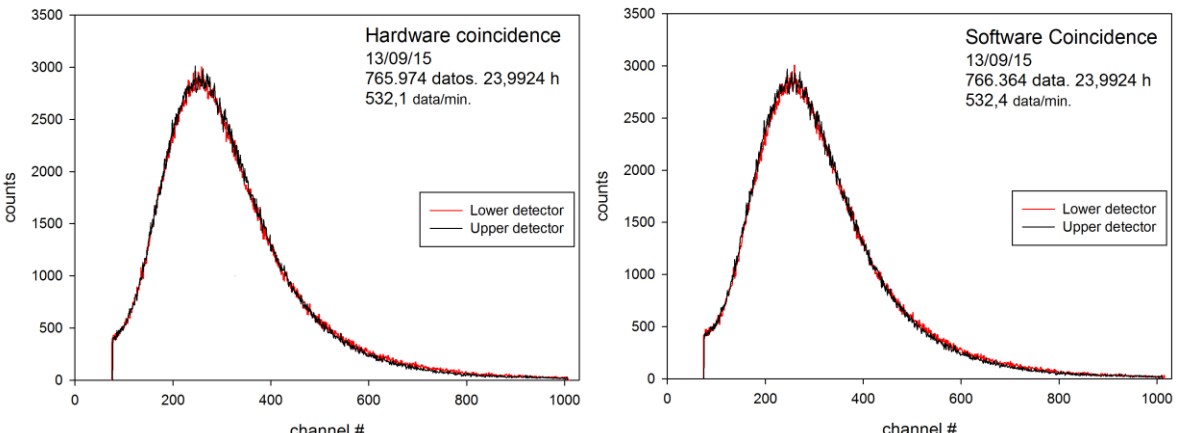

Figure 12. Comparative tests between data acquired with hardware (left) and software (right) coincidence by the muon telescope (MD1). Black and red lines correspond to the histogram of upper and lower detectors respectively.



| HARDWARE COINCIDENCE | | | | | SOFTWARE COINCIDENCE | | |
|---|---|---|---|---|---|---|---|
| ADC A | ADC B | INV A | INV B | dt | dt | ADC 1 | ADC 2 |
| 274 | 199 | 0 | 0 | 10 | 10 | 275 | 198 |
| 593 | 209 | 0 | 0 | 268 | 268 | 597 | 209 |
| 377 | 538 | 0 | 0 | 128 | 128 | 379 | 539 |
| 475 | 8191 | 0 | 1 | 159 | 160 | 478 | 270 |
| 217 | 285 | 0 | 0 | 2 | 1 | 218 | 285 |
| 350 | 301 | 0 | 0 | 2 | 2 | 352 | 301 |
| 125 | 277 | 0 | 0 | 91 | 91 | 125 | 277 |
| 286 | 222 | 0 | 0 | 15 | 15 | 287 | 221 |
| 327 | 250 | 0 | 0 | 13 | 13 | 328 | 249 |
| 346 | 227 | 0 | 0 | 24 | 24 | 347 | 226 |
| 335 | 596 | 0 | 0 | 4 | 5 | 337 | 598 |
| 383 | 276 | 0 | 0 | 61 | 60 | 385 | 276 |

ADC A & ADC B: Data from both ADCs in hardware coincidence. Range: 0 - 1023
ADC 1 & ADC 2: Data from both ADCs in software coincidence. Range: 0 – 1023
INV A & INV B: Invalid signal. 1 = active (conversion process error).
dt: time elapsed from previous event in ms.

Figure 13. Data analysis. We can see the same data acquired in hardware coincidence and software coincidence columns with an insignificant difference between values, which is due to ADC conversion. Sometimes, a conversion error is produced in an ADC, the invalid line is activated and the ADC data is out of range. That is what happens in the fourth row. In yellow, the invalid i2 activated (1) and the ADC2 data value (out of range = 8191). This is not a valid data. Although the hardware has detected the coincidence, this data is not registered in normal acquisition process because Data Ready signal is not activated. In this fragment would be one less data in hardware coincidence. This situation is inherent to ADC's operation and it has nothing to do with hardware or software coincidence.





Figure 14: From top to bottom, muon count-rate (black line) and smoothed count-rate (red line), neutron count-rate (black line) and smoothed count-rate (red line), solar wind density, solar wind temperature, solar wind speed, interplanetary magnetic field components and magnetic field intensity. Complex Ejecta refers to a complex solar wind structure composed by different interacting structures like shocks, ICMEs and interaction regions. The vertical purple lines mark the interplanetary shock positions and ejecta's limits.



(a)

(b)

(c)

Figure 15. Software coincidence with four PMTs (MD2). Three of them placed in the sides of a 100 cm x 100 cm x 5 cm scintillator. The fourth one is set with a 6 cm x 2 cm Bismute Germanate scintillator (BGO). Coincident particle tracks pass through a relative small region of the large scintillator, just under the BGO. When moving BGO through the big scintillator surface, the histograms shift depending on the distance and the angle formed by PMT axis and the line between PMT and scintillator impact place.

(a) Configuration sketch with BGO on the center of the big scintillator.

(b) Result histograms of upper-left configuration.

(c) Result histograms with different distances and angles.





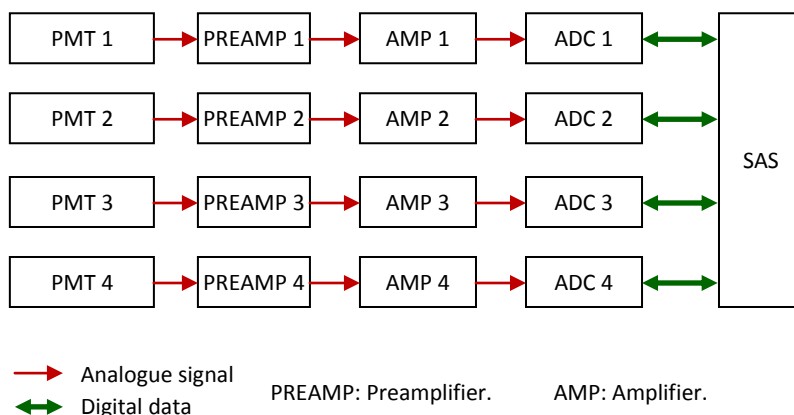

Figure 16. NIM modules and SAS interconnection block diagram working in software coincidence with four detection chains. The Signal from each PMT is amplified by preamplifier and amplifier in order to get the appropriate ADC input level. SAS detects coincidence and registers data.