# Peer review of "A coincidence detection system based on real-time software."

_Geoscientific Instrumentation, Methods and Data Systems, 2016_

## Referee Comment (RC1) · Anonymous Referee #1 · 7 Jul 2016

The MS No.: gi-2016-15 "A coincidence detection system based on real-time software" by Aysio et al. is logically composed and nicely graphically illustrated. A new system is based on the recurrent supervision of the Analog to-Digital Converters status. It has been tested on experimental muon measurements. This system works autonomously that, undoubtedly, provides definite advantages. Certainly, this MS will be useful for experts working in this field.

Remark: The authors note in the abstract that "Thus, our system replaces, at least, three Nuclear Instrument Modules (NIM) required by conventional coincidence systems, reducing its cost by a factor of 40 ...". However, this essential item practically do not explained in the MS. Obviously, at least a separate paragraph must be added.

I recommend to accept the MS to publication provided the above remark has been

taken into account

---

## Author Comment (AC1) · 9 Jul 2016

Thank you very much for your comments.

We agree. This item is not clearly explained through the MS.

Our SAS prototype has cost less than 150 €İt replaces, at least, three NIM modules when working in software coincidence detection: two SCAs and a coincidence detector module (see attached Fig.1). Moreover, it stores and transfers data to a PC, replacing another specific NIM interface module.

The cost of these four NIM modules is well above 6000 € (40 times the SAS cost).

(Please, see the attached supplement).

Author's changes in manuscript:

[Figure]

In order to clarify this item and according to the referee advice, the following changes have been made in the original manuscript:

1. The second paragraph in section 5 (lines 5 to 11, page 12) has been replaced with two new paragraphs.

2. The last paragraph in section 5 (lines 3 to 5 in page 13) has been removed.

3. A sentence has been added in figure 11 caption.

4. Three modules have been coloured in figure 11.

These changes will be included in final manuscript version.

The final aspect of section 5 and Figure 11, clarifying this topic, can be seen in attached supplement (in blue colour the text changes).

Please also note the supplement to this comment:
http://www.geosci-instrum-method-data-syst-discuss.net/gi-2016-15/gi-2016-15-AC1-supplement.pdf
* * *
[Figure]

[Figure]

Fig. 1. Schematic setup for hardware coincidence and software coincidence results comparison. We can see in yellow colour the unnecessary modules when SAS is working in software coincidence.

**Supplement:**

**5. SAS experimental validation.**

To validate the SAS reliability and proper functioning, several data acquisition experiments with the muon telescope MD1 (see Sect. 2.1) were performed. Both, hardware and software coincidence configurations were simultaneously tested and their results were compared.

Figure 11 shows the experimental setup operating in hardware (blue background block) and software (red background block) coincidence simultaneously. In hardware coincidence, the SAS is used to store data, so it does not work as a coincidence detector module and that is the reason why its software has been slightly simplified; it waits for the activation of both DR signals to read and store data, and then resets the ADCs.

The software coincidence block shows how, this configuration, is significantly simpler than the one based on hardware coincidence, saving three modules: two SCAs and one coincidence detector module (yellow modules in Fig. 11). Moreover, the SAS has the capability of storing and transferring data to a PC, avoiding the use of an interface module. From an economical point of view, the total cost of those four NIM modules is well above 6000 € (only one SCA module costs more than 1600 €) and the SAS implementation components have cost less than 150 €. So, we can say that the SAS, working in software coincidence, reduces the value of the laboratory equipment replaced by a factor of 40.

To make comparisons between both types of coincidence detection systems, we acquired data during one day with the experimental setup shown in Fig. 11. Obviously, the data registered by both software and hardware coincidence chains should be identical. Figure 12 shows the corresponding histograms produced, which are nearly identical, showing only a minor difference in the total amount of data acquired by both systems (0.05 %).

Although this difference can be considered negligible, further tests were performed in order to find out the origin of this discrepancy. Sometimes, the ADC conversion process produces errors and conversion is aborted (see Sect. 2.4). In these cases, DR signal is not generated and INV signal activated, which causes data is not registered. Ad-hoc code has been written to register INV signal and several samples has been taken and analyzed. As it can be seen in Fig. 13, an error causes the INV

activation in hardware coincidence chain. However the software coincidence prototype stores the correct value because its ADCs have not produced conversion errors. In normal operation, this hardware coincidence data would not be registered and, as a result, the total amount of hardware coincidence data would be lower than the one registered with software coincidence. That is the origin of the small difference between the histograms corresponding to hardware and software coincidence shown in Fig. 12. Therefore, seeing how the rest of data are similar in time and amplitude values, we conclude that under the experimental conditions used in this work, both kinds of coincidence detection systems (hardware and software) produce equivalent result.

[Figure]

PREAMP: Preamplifier.    AMP: Amplifier.    SCA: Single Channel Analyzer.    PMT: Photomultiplier.

Figure 11. Schematic setup for hardware coincidence and software coincidence results comparison. The
same analogue signals detected by PMT1 and PMT2 are introduced into both hardware coincidence and
software coincidence chains. Working in hardware coincidence, SAS 1 only stores data from both
chains. Working in software coincidence, SAS 2 detects coincidence and stores data. We can see in
yellow colour the unnecessary modules when SAS is working in software coincidence.

---

## Referee Comment (RC2) · Anonymous Referee #2 · 14 Jul 2016

[referee-annotated manuscript omitted]

---

## Author Comment (AC2) · 18 Jul 2016

Author's response:

Thank you very much for the comments. They helped us to clarify several important issues and significantly improve the quality of our manuscript.

Please, see attached document with changes in blue.

1. (p4 l5) Authors have to justify the choice of voltage of operation based on the experimentally determined PMT response.

We agree. A new figure 2 and the following paragraph have been added in page 4.

"We can see in Fig. 2 the variation of the counter response with the bias voltage for both PMTs used in the experiment. Bearing in mind they work in coincidence coupled

to identical scintillators, we have chosen 1270 V as optimal value because it is the point where both PMTs have the same response within the plateau."

2. (p4 l14) A photo of this device is necessary

We agree. A photo has been added (new figure 3).

3. (p5 l25) This paragraph would be better understood if diagrams are used

We agree. Diagrams have been included (new figure 5).

4. (p6 l25) A full list of the physical information that may be obtained should be given.

It is difficult to make a complete list of capabilities and uses of coincidence-based systems to obtain physical information because of the large number of them. On the other hand, the reason to include this sentence in the manuscript was only to stress the importance of coincidence method giving some example of use. In order to support this claim, we have extended and referenced the initial list.

"...may provide relevant physical information such as particle identification by the use of dE vs dE or dE vs E techniques (del Peral et al, 1995) or by means of some shielding block between pilled detectors (Chilingarian et al, 2009a); particle impact point on the detector surface (Hasebe et al, 1988) and particle energy deposition in detectors or incident direction (Karapetyan et al, 2013). Moreover, coincidence systems are used in different research areas such as medical applications, quantum physics or optics (Joost and Salomon, 2015)."

5. (p10 l1) Why?

We have added a new table (table 2) in order to justify the use of 37 GPIO.

6. (p10 l6) Figure 9 is unnecessary.

We agree. Figure 9 has been removed.

7. (p14 l17) Magnetic field rotations may only be seen when components are shown.

These do not appear in Fig 14.

Magnetic field rotation already was shown in figure 14. Please, see from bottom to top the second row. Nevertheless, the figure has been adapted to point out the magnetic field rotations by shadowing the adequate region. In addition, a new shock line and the probably location of the associated magnetic cloud have been marked in the updated figure (now, Fig. 16). The text has been slightly reworded according to the new figure.

8. (p14 l23) An estimate of the average response of both detectors and the reasons for that should be given. The decrease in the muon telescope is much faster than that of the NM. Authors must try to give an explanation for this.

We don't know what exactly the referee means with average response. We have explained at the beginning of this section that, because of CaLMa cut-off rigidity, the energy of primary CR that produce neutrons measured in CaLMa is higher than 7 GeV when protons are the primary CR. The CaLMa background count rate is 72.62 Hz. For the muon telescope, the expected energy for primary CR is higher than 10 GeV and its background count-rate is 7.66 Hz. Regarding to the second part of the referee's comment, in our opinion, what it is observed in figure 14 is just the opposite, a sharp decrease in neutron and a softer decrease in muons, as could be expected for primary cosmic rays with a higher energy threshold. Nevertheless, in order to clarify it, we have re-written the last paragraph in section 6.1.

"The good agreement between the muon and neutron data, presented in Fig. 16, validates the software-based coincidence system used to acquire the muon data. Both of them show a clear response to the passage of interplanetary disturbances. The difference in their count-rates decreases observed by both instruments in shape (faster, sharper and deeper decrease in CaLMa) and in magnitude is likely related to the different energy of the primary cosmic ray producing the secondary neutrons and muons observed at ground level, as could be expected when the primary cosmic ray energy threshold for CaLMa (neutrons mainly) is about 7 GeV and for the muon telescope is

about 10 GeV."

9. Finally, all English language mistakes have been corrected as suggested.

Author's changes in manuscript.

In order to improve the manuscript comprehension and according to the referee's advice, the following changes have been made in the original manuscript:

1. A new figure 2 has been added to clarify the bias voltage chosen for PMTs used in muon detector 1 (MD 1).

2. A new photo (figure 3) has been added showing the scintillator and the BGO in section 2.1.

3. A new figure 5 has been included in order to clarify ADC conversion process.

4. First paragraph in section 3 has been modified and four references added.

5. Table 2 has been added to justify the use of 37 GPIOs.

6. Figure 9 has been removed.

7. Figure 14 has been changed and second paragraph in page 14 has been rewritten.

8. Third paragraph in page 14 has been rewritten.

9. All suggested English language corrections have been made.

These changes will be included in final manuscript version. The final aspect can be seen in attached document. The text changes coloured in blue.

Please also note the supplement to this comment:
http://www.geosci-instrum-method-data-syst-discuss.net/gi-2016-15/gi-2016-15-AC2-supplement.pdf

15, 2016.

Table 2. GPIOs required by the processor card to control the ADC. We need 18 lines to read data and control one ADC. Another line is needed to enable and disable the interface buffers, which has been added to protect the processor card.

| Name | Acro. | # Lines | Description |
|---|---|---|---|
| Data | D | 13 | Binary data value |
| Data Ready | DR | 1 | Active when conversion is complete. |
| Invalid | INV | 1 | Conversion error |
| Overflow | OVF | 1 | Value exceeds ULD settings |
| Enable Data | ED | 1 | Gate the 13-bit data onto the output lines |
| Data Accepted | DA | 1 | Acknowledgment of data acceptance. Reset the ADC |
| TOTAL lines 1 ADC | | 18 | |
| Line to enable interface buffers | | 1 | |
| TOTAL to control 2 ADCs | | 37 | 18+18+1 |

**Fig. 1.** Table 2

[Figure]

Figure 2. Counting rate versus voltage for PMTs type 53AVP (Philips). Note the plateau below 1500 V and the crossing point of both curves in 1270 V (bias voltage chosen).

**Fig. 2.** Figure 2

[Figure]

Figure 3. Main scintillator (100 cm x 100 cm x 5 cm). Four PMTs inside its pyramidal guides gather the light from lateral sides of scintillator. The BGO inside the little black box can be moved over the scintillator surface. The BGO and three PMTs work in coincidence, thus only the muon trajectories crossing the BGO will be detected and the amplitude of PMTs pulses will carry position information. The BGO is used to calibrate the system. All system is located inside a closed dark chamber.

**Fig. 3.** Figure 3

[Figure]

Figure 5. ADC conversion process. The ADC detects a peak when the input signal rises above the LLD threshold and below the ULD threshold. The detection process ends when the input pulse falls below 90% of its peak amplitude. In that moment, the signal input is disabled and the conversion process starts. If an error occurs in the conversion process, DR signal remains inactive and input signal enabled again. If conversion process is OK, DR is activated, the Data Acquisition System reads the data and actives DA signal, which causes the DA to go to inactive and the signal input to be enabled again.

**Fig. 4.** Figure 5

[Figure]

**Fig. 5.** Figure 16

**Supplement:**

**A coincidence detection system based on real-time software.**

Sindulfo Ayuso1, Juan José Blanco1,2 José Medina1, Raúl Gómez-Herrero1,2, Oscar García-Población1,3, Ignacio García Tejedor1,3.

[1]{Castilla-La Mancha Neutron Monitor. Space Research Group. Parque Científico y Tecnológico de Castilla-La Mancha. Avda. Buendía, 11. 19005 Guadalajara, Spain.}

[2]{Physics Department. Space Research Group. Universidad de Alcalá. Ctra. Madrid-Barcelona km33,6. 28871 Alcalá de Henares. Spain.}

[revised manuscript text omitted]

electronic devices like one or more ADCs.

10

15

The goals of this work are, firstly, the establishment of the theoretical background and conditions allowing software-based real-time coincidence detection (Sect. 3); secondly, the prototype implementation with a low-cost development platform and minimal and simple hardware and software designs (Sect. 4); thirdly, the validation of operation extracting data from a muon telescope (Sect. 5) and, finally, the prototype testing in two practical applications (Sect. 6). In addition, we will see how our prototype is able to replace at least three NIM modules used in conventional setup for coincidence detection.

**2 Experimental setup.**

20 In this section we describe the different elements that have been used in our experiment, mainly two muon detectors and some NIM modules, and how they have been setup to achieve the results presented in this paper.

**2.1 Muon detectors.**

We have used two different muon detection systems based on plastic scintillators. The first device

[revised manuscript text omitted]

**20 3 Coincidence**

25

Particle detection systems are often based in multiple detection layers operating in coincidence. These coincidence-based systems may provide relevant physical information such as particle identification by the use of dE vs dE or dE vs E techniques (del Peral et al, 1995) or by means of some shielding block between pilled detectors (Chilingarian et al, 2009a); particle impact point on the detector surface (Hasebe et al, 1988) and particle energy deposition in detectors or incident direction (Karapetyan et al,

2013). Moreover, coincidence systems are used in different research areas such as medical applications, quantum physics or optics (Joost and Salomon, 2015).

[revised manuscript text omitted]

temperature as can be expected when a well-developed magnetic cloud is observed in the solar wind (first shadowed region in Fig. 16). A second rotation in magnetic field components is observed between 22/12/1014 17:00 and 23/12/2014, (second shadowed region in Fig. 16) suggesting the presence of a second ICME, however solar wind properties show less clear signatures (Hidalgo, 2013).

- 5 The good agreement between the muon and neutron data, presented in Fig. 16, validates the softwarebased coincidence system used to acquire the muon data. Both of them show a clear response to the passage of interplanetary disturbances. The difference in their count-rates decreases observed by both instruments in shape (faster, sharper and deeper decrease in CaLMa) and in magnitude is likely related to the different energy of the primary cosmic ray producing the secondary neutrons and muons observed
- 10 at ground level, as could be expected when the primary cosmic ray energy threshold for CaLMa (neutrons mainly) is about 7 GeV and for the muon telescope is about 10 GeV.

**6.2 Position-sensitive muon detector.**

[revised manuscript text omitted]